# Computerized history-taking improves data quality for clinical decision-making— Comparison of EHR and computer-acquired history data in patients with chest pain

**David Zakim**[1]*, **Helge Brandberg**[2], **Sami El Amrani**[1‡], **Andreas Hultgren**[1‡], **Natalia Stathakarou**[1‡], **Sokratis Nifakos**[1‡], **Thomas Kahan**[2], **Jonas Spaak**[2], **Sabine Koch**[1], **Carl Johan Sundberg**[1,3]

1 Department of Learning, Informatics, Management and Ethics, Medical Management Centre, and Health Informatics Centre, Karolinska Institutet, Stockholm, Sweden, 2 Division of Cardiovascular Medicine, Department of Clinical Sciences, Danderyd Hospital, Karolinska Institutet, Solna, Stockholm County, Sweden, 3 Department of Physiology and Pharmacology, Karolinska Institutet, Stockholm, Sweden

☯ These authors contributed equally to this work.
‡ These authors also contributed equally to this work.
* david.zakim@ki.se

**Data Availability Statement:** Data cannot be shared publicly because of ethical restrictions from the Swedish authorities as the data contain

## Abstract

Patients' medical histories are the salient dataset for diagnosis. Prior work shows consistently, however, that medical history-taking by physicians generally is incomplete and not accurate. Such findings suggest that methods to improve the completeness and accuracy of medical history data could have clinical value. We address this issue with expert system software to enable automated history-taking by computers interacting directly with patients, *i.e.* computerized history-taking (CHT). Here we compare the completeness and accuracy of medical history data collected and recorded by physicians in electronic health records (EHR) with data collected by CHT for patients presenting to an emergency room with acute chest pain. Physician history-taking and CHT occurred at the same ED visit for all patients. CHT almost always preceded examination by a physician. Data fields analyzed were relevant to the differential diagnosis of chest pain and comprised information obtainable only by interviewing patients. Measures of data quality were completeness and consistency of negative and positive findings in EHR as compared with CHT datasets. Data significant for the differential of chest pain was missing randomly in all EHRs across all data items analyzed so that the dimensionality of EHR data was limited. CHT files were near complete for all data elements reviewed. Separate from the incompleteness of EHR data, there were frequent factual inconsistencies between EHR and CHT data across all data elements. EHR data did not contain representations of symptoms that were consistent with those reported by patients during CHT.

**Trial registration:** This study is registered at https://www.clinicaltrials.gov (unique identifier: NCT03439449).

potentially identifying and sensitive patient information. Data could however be available for researchers who meet the criteria for access to confidential data, upon reasonable request to the authors and with permission of the Swedish Ethical Review Authority (https://etikprovningsmyndigheten.se registrator@etikprovning.se).

**Funding:** This work was funded by the Robert Bosch Stiftung (https://www.bosch-stiftung.de/de, Stuttgart, Germany), grant number 11.5.1000.0258.0 to DZ. Region Stockholm (ALF project; https://www.vr.se/english/about-us/organisation/advisory-groups-and-administrative-offices/office-for-alf.html, Stockholm, Sweden), grant number 20190593 to TK. Karolinska Institutet Research Foundation (https://staff.ki.se/ki-research-foundation-grants-2020-2021, Stockholm, Sweden) and Stiftelsen Hjärtat (http://www.stiftelsenhjartat.se, Stockholm, Sweden) to TK. Funders had no role or influence on the design and conduct of the research, including software development, and were not involved in data analysis, conclusions drawn from the data, and drafting or editing the manuscript.

**Competing interests:** All authors have completed the ICMJE uniform disclosure form at www.icmje.org/coi_disclosure.pdf and declare: no support from any organization for the submitted work; no financial relationships with any organizations that might have an interest in the submitted work in the previous three years; no other relationships or activities that could appear to have influenced the submitted work. DZ is the sole inventor on US Patent 7,379,885: "A system and method for obtaining, processing and evaluating patient information for diagnosing disease and selecting treatment." This patent has not been maintained and is no longer valid. All copyrights to CLEOS technology, language, images, and knowledge content have been assigned, to Karolinska Institutet, Stockholm, Sweden without compensation and royalty rights to DZ. Karolinska Institutet is a public university and the sole owner of the CLEOS program. Apart from Karolinska Institutet and its subsidiaries, no individuals or companies may be owners or receive royalties or other revenue from use of CLEOS technology, language, images, knowledge content, data, or clinical insights and/or computer algorithms generated from analysis of data acquired by the program. The existing patent to DZ does not affect our adherence to PLOS ONE policies on sharing data.

## Introduction

Medical histories are the salient datasets for characterizing clinical states and informing diagnostic and treatment decisions [1–5]. Whenever studied, however, medical history data recorded by physicians is found consistently to be incomplete, inaccurate, biased, and not always fact-based [6–16]. These inadequacies may reflect, in part, poor documentation of what the physician knows about the patient. But real-time observations reveal too that there are deficiencies in the history-taking process as such [17, 18]. And since inadequate history data can lead to diagnostic errors [1, 19–21], it appears that we need better methods for history-taking than physicians interviewing patients.

The key barriers to effective history-taking by physicians are insufficient time with patients and the enormity of the knowledge for history-taking [22, 23]. It was realized even at the beginning of the computer-age that expert system software could solve these problems [24, 25]. Yet today, there is scant academic interest in developing expert systems for computerized history taking (CHT), *i.e.* automated, dynamic history taking as a patient interacts with a computer [26]. We bring attention to the possible value of CHT in the present work. We compare medical history data collected and entered into EHRs by physicians with history data collected by CHT at the same ED visit from 410 patients presenting to an emergency department (ED) with acute chest pain.

## Materials and methods

### Experimental design

The results we report are from a within person study in which the experimental intervention was deployment of two different methods for collecting and storing medical history data. The first of these was physicians providing routine ED care and entering their findings into an electronic health record (EHR). The second was expert system software trained to interview patients with acute chest pain, *i.e.* CHT. Data entries were stored automatically by the CHT program. Every enrolled patient was interviewed by a physician and the expert system software during the same ED visit. Data collected by CHT was not available to ED staff.

### Study setting and recruitment of patients

All patients in this study presented to the ED of Danderyd University Hospital, Stockholm, Sweden with a recorded chief complaint of "chest pain" at entry to the ED. Patients with chest pain as presenting complaint were recruited consecutively if they were 18 years or older, had a non-diagnostic ECG for acute myocardial infarction or angina, had native fluency in Swedish or English, and had an equivalent Manchester triage score of 3, 4 or 5. Some eligible patients were excluded because they arrived at the ED without eyeglasses and were unable to read the screen of a tablet computing device. There were no other criteria for inclusion or exclusion. Patients were recruited between October 2017 and November 2018. Participation was skewed toward male patients (57% of 410 patients) as compared with a distribution of 50.4% male patients in the Danderyd ED population of patients with chest pain [27]. The age distributions of male and female patients, categorized by cut points in the HEART score [28], were essentially identical (Table 1). Swedish law does not allow questioning patients about ethnic origin.

Patients were recruited by term 9 Karolinska Institutet medical students (SEA and AH), who were familiar with the CLEOS program and the purposes of the research plan. SEA and AH explained to patients the purpose of the research program and demonstrated how to interact with the CHT program running on an iPad® (Apple Inc, Cupertino, CA, USA). Patients were told that agreeing or declining to participate would not affect care, would not affect wait

**Table 1. Gender and age distributions of enrolled patients.**

| Gender | Male | Female |
|---|---|---|
| Number enrolled | 236 | 174 |
| Fraction ages 18–44 | 0.311 | 0.302 |
| Fraction ages 45–64 | 0.408 | 0.409 |
| Fraction age $\geq$ 65 | 0.281 | 0.298 |

time in the emergency department, would not prolong the duration of their stay in the emergency department, and would not affect eventual discharge to home or admission. Patients were informed they were free to end the computerized interview at their discretion and that the computerized interview would end at the time a decision was made to discharge or admit whether or not the computerized interview was complete. Patients were informed that the data collected by the computerized interview would be unavailable to their providers of care in the ED and that the data collected would be used exclusively for clinical research. Patients also were informed that the identifiers of the data collected by the CHT program were stored in a locked safe at Danderyd University Hospital and that the only people with access to the safe would be Karolinska Institute faculty participating in the research program. SEA and AH revisited participating patients during CHT sessions to insure patients were able to navigate the program on the iPAD.

## CHT software

The CHT software (CLEOS) is described in detail elsewhere [23, 26, 29]. In brief, CLEOS is a dynamic expert system for history-taking from adults with acute and long-term complications from chronic disease. The knowledge base emulates the thinking of expert physicians to pose questions that are relevant to the program's working differential diagnosis, which is formulated first from the patient's chief complaint and demographic profile. Data collected is interpreted continuously in the context of applicable pathophysiology to direct questioning to resolving the working differential diagnosis. Review of systems data is collected by the same scheme. The knowledge base of questions and answers is developed in English comprehensible to a native English-speaker with a U.S. sixth grade education. Textual questions have answers sets of Yes/No, Yes/No/Uncertain, or multiple choice answers. Questions are graphic whenever this is possible, as for example when the site of primary or radiated pain is queried. Each answer is coded to indicate the question asked and the answer entered. Patients' answers are stored as codes. The English language version of CLEOS was translated to Swedish by professional translators experienced with technical translation. The Swedish translation was tested for clinical accuracy by Karolinska Institute physician and nursing staffs for comprehension by a representative, demographic of the Swedish population. Problematic language was corrected before deployment in the work described.

This study used a version of CLEOS with 17,500 decision nodes, more than 10,000 pages of queries and several thousand rules that continuously determine the next most appropriate question, patient-by-patient, by interpreting the clinical significance of data as it is collected. The CHT interview ran on a server at Karolinska Institutet connected via VPN to the ED. CLEOS is owned fully by Karolinska Institutet, a public university. The standard CLEOS interview, which begins by querying the chief complaint, was modified for the present work. Instead of asking for the chief complaint, patients were asked by the question in Fig 1 to confirm or deny that chest pain was their chief complaint. All patients were interviewed by the Swedish language version of CLEOS.

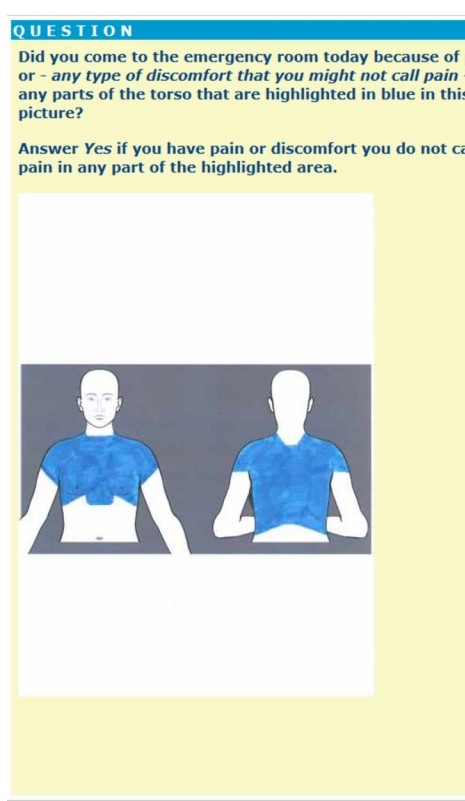

**Fig 1. Image presented to patients to confirm that chest pain was the chief complaint.** Patients were asked to confirm with a "yes" or "no" that they were seeking medical care because of pain somewhere in the blue region of the image.

### Time-relationship between physician history-taking and CHT interview

Patients interacted with the CHT software during wait times. CHT was interrupted at patient request or for care. Almost all patients began CHT prior to examination by their ED physician. We have exact time-stamps for the start of a CHT session and for interruptions during a session. There was no reliable data for the exact time of physician examinations, however. Ten of the 410 patients for whom data is reviewed arrived at the ED in early hours of the morning when medical students were not present to recruit patients. These 10 patients were examined by a physician prior to the start of a CHT interview. Search of time stamps in CHT records revealed that 20 of 400 patients arriving in the ED during hours of active recruitment had at least one interruption of 10 min. or longer. A first history-taking session by a physician might enhance patient recall of information during a subsequent CHT session and might have occurred or began during a CHT interview with a long delay between answers for successive questions. We thus compared data in EHRs and CHT files for those CHTs that began after a physician's examination (10 patients) and those with a greater than 10 min interruption (20 patients). Differences between EHR and CHT data for these 30 patients were not different qualitatively or quantitatively from comparisons across the other 380 patients. Data analyses are presented for a single group of 410 patients.

### Data extraction

EHR records corresponding to personal national identity numbers were accessed at the Danderyd University Hospital ED. Personal national identity numbers were not copied or stored.

The date of signed patient consent was used to verify that EHR and CHT data were for the same ED visit. Data elements analyzed in EHR and CHT files were clinical attributes relevant to the differential diagnosis of chest pain and known only to the patient, e.g., site(s) of chest pain, time of onset, setting for onset, frequency of episodic pain, constancy of pain, and so on. Data was extracted from EHRs by SEA and AH. Extracted datasets were identified only by an accession number corresponding to a CLEOS file. CLEOS codes representing specific answers to specific questions were extracted by DZ, who had no knowledge of patient identifiers. The chi-square statistic for categorical variables was used to determine the significance of differences between data elements in EHR and CHT data sets. To calculate $X^2_1$, the number of patients in EHR data with a given finding was the observed value; the number in CHT data with the given finding was the expected value. P-values for level of significance were determined for $X^2$ for 1 degree of freedom.

## Exclusion of selected data sets

We found that repetitive tapping by patients on the button sending answers from iPAD to server and requesting a next question corrupted the pathway of knowledge graphs. The problem was corrected during the course of this study. Interviews corrupted in this way were excluded from analysis. We excluded data for four patients, who did not confirm chest pain as chief complaint. We excluded data for two additional patients, who appeared to make deliberately false entries during CHT. The evidence for this was large discrepancies between year of birth in the CHT record (selected from a drop-down menu) and age recorded in the patient's EHR, apparent indiscriminate selection of all answers for sites of pain and sites of radiated pain and answers to questions that were not consistent with each other. The remaining 410 patients were included in the study.

The protocol was approved by the Stockholm Regional Ethical Committee (now Swedish Ethical Review Authority) (No 2015/1955-31).

## Results

### Data for location of primary pain

Of the 410 EHRs, 213 had no entry for a specific site of chest pain or an entry too imprecise to assign pain to an anatomic location in the chest, shoulders or epigastrium. All 410 CHT records had a primary site of pain entered by patients interacting with a version of Fig 2 without numbering of the anatomic areas.

S1 Fig displays EHR and CHT data for location(s) of primary pain for all 410 patients. Each line in S1 Fig shows locations of pain for a single patient with EHR findings in the upper triangle and CHT findings in the lower triangle of each numbered rectangle. S1 Fig shows that CHT datasets had more precise locations of pain and a far greater heterogeneity of pain patterns. S2 Fig displays examples of EHR and CHT data for locations of pain projected onto the chest.

We analyzed EHRs and CHTs specifically for entries for central chest pain (Table 2). Congruence of EHR and CHT data for central chest pain was limited. Of 256 patients with central pain during CHT, EHR data for pain location was missing in 120. CHT reports for the presence and absence of central pain did not confirm frequent EHR entries for the presence of central pain or the absence of central pain.

We searched for evidence that discrepancies between EHRs and CHTs for locations of pain reflected inaccurate recall of pain that remitted. But discrepancies between EHR and CHT data for pain location were equally frequent for patients with or without pain during CHT. CHT data showed too that pain patterns were stable over time. Patients with a first onset of

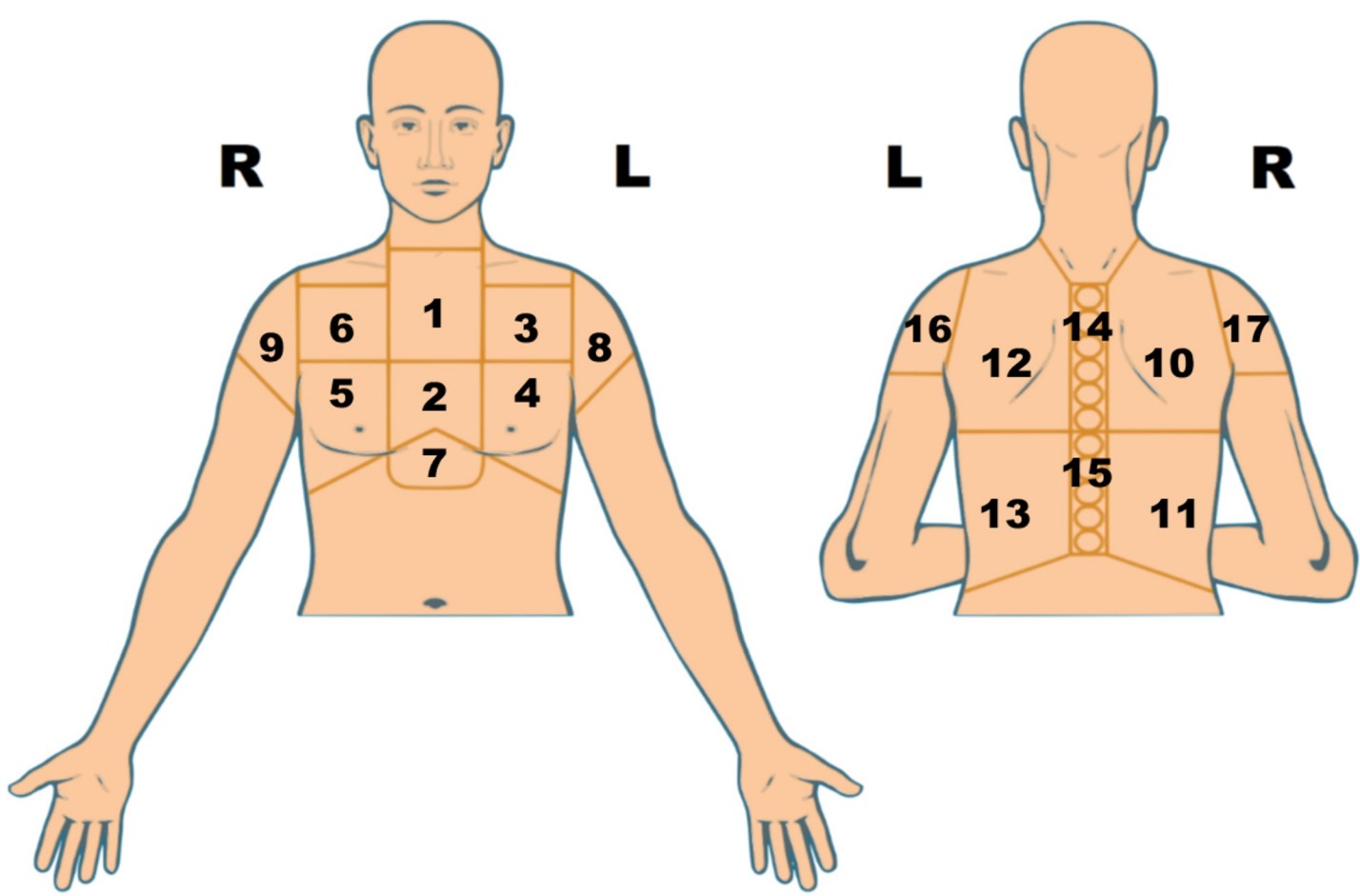

**Fig 2. Image used by patients to enter sites of chest pain.** Tapping on any region of the image added blue color to it. The patient could deselect a blue area with a second tap. The areas in the image presented to patients were not numbered. We show numbers here because they indicate the areas of pain in S1 Fig.

pain days to weeks before presentation and negative self-reported histories for angina, myocardial infarction and revascularization reported that sites of pain did not change in the intervals between a first occurrence and the presenting event. CHT data indicated that pain at presentation was the same as in prior episodes of angina or infarction for all but one patient with a CHT reported history of angina or infarction.

## Location of radiated pain

Positive or negative entries for radiated pain were missing in 224 of 410 EHRs. An entry for radiated pain was missing in 2 of 410 CHT files because of early termination of the interview.

**Table 2. Findings for central chest pain in EHR and CHT records.**

| Site of Chest Pain | EHR Data Patients | CHT Data Patients | p-Value EHR *vs* CHT | Congruence EHR and CHT Findings Patients (%) |
|---|---|---|---|---|
| Central Chest Pain | 94 | 256 | < 0.001 | 77 (30.1) |
| Central Chest Pain Exclusively | 88 | 62 | < 0.001 | 30 (48.4) |
| No Central Chest Pain | 92 | 154 | < 0.001 | 41 (26.6) |

EHR and CHT data was reviewed for 410 patients. An EHR and CHT record was available for every patient. Congruence between EHR and CHT findings is the instances in which the same patient was identified by EHR and CHT data. P-values are from a $X^2$ distribution table for one degree of freedom and $X^2$ calculated as in Methods. The denominator for percent congruence was the number of patients identified by CHT data.

**Table 3. Patterns of radiated pain.**

| Site Radiated Pain | EHR Data Patients | CHT Data Patients | p-Value EHR *vs* CHT | Congruence EHR and CHT Data Patients (%) |
|---|---|---|---|---|
| R + L shoulder or R + L arm | 8 | 63 | < 0.001 | 3 (4.8%) |
| R shoulder, arm or hand; no left-sided radiation | 1 | 24 | < 0.001 | 1 (4.1%) |
| Jaw and/or neck | 30 | 48 | < 0.001 | 6 (12.%) |

EHR and CHT data was reviewed for 410 patients. Patients with radiation to R and L shoulders and radiated pain to an arm are not identified separately. P-values are from a $X^2$ distribution table for one degree of freedom and $X^2$ calculated as in Methods. The denominator for percent congruence was the number of patients identified in CHT files.

CHT data for radiated pain was missing for 6 patients with no entries of anterior chest pain. This occurred because CLEOS was programmed in error to not ask about radiated pain in the absence of anterior chest pain. We searched EHR and CHT data specifically for patterns of radiated pain relevant to the differential of chest pain (Table 3) [28]. CHT as compared with EHR data identified significantly more patients with relevant radiation patterns (Table 3). CHT data did not confirm most positive EHR entries for these patterns of pain. And as noted in Table 3, 80% of patients with jaw/neck radiation in EHR data denied this finding during CHT. We also found that 50% of explicitly negative EHR entries for radiation to the neck/jaw were for patients reporting radiation of pain to the jaw and/or neck during CHT.

## Setting for the onset of pain

Table 4 summarizes findings for onset of pain during physical exercise or emotional upset. EHR data was absent or not interpretable for the setting of onset of pain for 30 of 44 patients with CHT data for onset of pain with exercise. Eighty percent of patients identified in EHR data with exercise-induced onset of pain denied during CHT that this was a correct representation of the setting in which their pain began. EHR data was even less consistent with CHT data for patients reporting onset of pain during emotional upset. Premature termination of CHT led to missing CHT data for one patient with EHR data for onset of pain during physical activity.

## Frequency and duration of pain

EHR data was missing for 81 patients reporting two or more episodes of pain in the 24 hours before presentation, during CHT; 8 of 53 patients cited in EHRs with two or more episodes of pain reported constant pain during CHT (Table 5). There was no significant difference for EHR and CHT data for < 2 episodes of pain. But lack of congruence between EHR and CHT data for < 2 episodes of pain was significant with a p-value of <0.001, Data for the frequency

**Table 4. EHR and CHT data for settings for onset of pain.**

| Setting for Onset of Pain | EHR Data Patients | CHT Data Patients | p-Value EHR *vs* CHT | Congruence EHR and CHT Data Patients (%) |
|---|---|---|---|---|
| Physical activity | 68 | 44 | < 0.001 | 14 (31.8) |
| Not physical activity | 153 | 360 | < 0.001 | 142 (39.4) |
| Emotional upset | 7 | 36 | < 0.001 | 6 (16.0) |
| Not physical activity; not emotional upset | 3 | 324 | < 0.001 | 3 (< 1) |

EHR and CHT data was reviewed for 410 patients.

P-values are from a $X^2$ distribution table for one degree of freedom and $X^2$ calculated as in Methods. The denominator for percent congruence was the number of patients identified in CHT files.

**Table 5. Frequency and duration of pain.**

| Attribute | EHR Data Patients | CHT Data Patients | p-Value EHR *vs* CHT | Congruence EHR and CHT Data Patients (%) |
|---|---|---|---|---|
| 2 or More Episodes of pain in prior 24h | 53 | 123 | < 0.001 | 28 (22.7) |
| Less than 2 Episodes of Pain in prior 24h | 106 | 102 | NS | 27 (26.5) |
| Constant Pain from Onset or Pain Became Constant | 47 | 119 | < 0.001 | 24 (20.1) |

P-values are from a $X^2$ distribution table for one degree of freedom and $X^2$ calculated as in Methods. NS means not significant. The denominator for percent congruence was the number of patients identified by CHT data.

of pain was missing in 7 CHTs that ended prematurely for uncertain reasons. CHT data for frequency of pain was missing in one patient with epigastric and right posterior chest pain at the level of T7-12. The CHT program excluded acute coronary syndrome from the working differential for this patient.

EHR data was missing for 95 of 119 patients with CHT findings of constant pain since onset or pain that became constant. Time domain data was missing in EHRs for 22 patients reporting constant pain for 12–24 hrs during CHT and for 49 patients reporting constant pain for > 24 hrs during CHT.

## Data for associated symptoms

EHR data was missing for a significant proportion of patients reporting sweating during CHT; and there was poor congruence between EHRs and CHTs for explicitly positive and negative findings for sweating with chest pain (Table 6). There was no significant difference between EHR and CHT data for the number of patients with no sweating.

But the lack of congruence between EHRs and CHTs for absence of sweating was significant at p <0.001. The lack of congruence between EHR and CHT data for pain exacerbated by respiration also was significant (p < 0.001). EHRs cited that some patients had pain with breathing and others had pain with inspiration. CLEOS asked patients with a pleuritic component of pain whether pain was exacerbated by breathing or caused by inspiration and absent during breath-holding. CHT data identified 9 patients with pain relieved by breath- holding. None of these patients was identified by EHR data. Nine patients cited in EHRs with pain during inspiration had a pleural component of pain in CHT data but indicated that breathing exacerbated pain that was not relieved by breath-holding.

**Table 6. Associated symptoms of sweating and pleuritic pain.**

| Attribute | EHR Data Patients | CHT Data Patients | p-Value EHR *vs* CHT | Congruence EHR and CHT Data Patients (%) |
|---|---|---|---|---|
| Sweating by history | 23 | 106 | < 0.001 | 14 (13%) |
| Negative sweating by history | 243 | 263 | NS | 187 (71%) |
| Pain worse with breathing | 64 | 60 | NS | 35 (58%) |
| Pain only with inspiration | 0 | 9 | < 0.001 | 0 |

EHR data for patients with pain caused by inspiration did not specify relief by breath-holding. CHT data for pain only on inspiration means pain was absent during breath-holding. CHT data for pain only on inspiration is included in patients with pain exacerbated by breathing. Ten percent of CHT interviews ended before associated symptoms were queried. P-values are from a $X^2$ distribution table for one degree of freedom and $X^2$ calculated as in Methods. NS means not significant. The denominator for percent congruence was the numbers of patients identified by CHT data.

**Table 7. Use and effect of self-administered nitroglycerin.**

| Attribute | CHT Data Patients | EHR Data Patients | p-Value EHR *vs* CHT | Congruence EHR and CHT Data Patients (%) |
|---|---|---|---|---|
| Self-administered nitroglycerin | 26 | 16 | 0.05 | NA |
| Complete pain relief by nitroglycerin | 10 | 6 | 0.2 | 6 (60%) |
| No pain relief after nitroglycerin | 16 | 10 | 0.2 | 8 (50%) |

EHR and CHT data was reviewed for 410 patients. Data is keyed to the CHT finding of self-administered nitroglycerin prior to medical care. EHRs were searched for an EHR entry of nitroglycerin use only for patients with a CHT entry for self-administered nitroglycerin. Patients in the row Self-administered nitro-glycerin and column EHR Data are the number of EHRs with entries for use of nitroglycerin. CHT data indicated that 2 patients had complete pain-relief only between 15 and 30 minutes after self-administration of nitroglycerin. These patients are listed as no relief from nitroglycerin. NA indicates not applicable. P-values are from a $X^2$ distribution table for one degree of freedom and $X^2$ calculated as in Methods.

## Use of nitroglycerin

CHT interview identified 26 patients, who self-administered nitroglycerin prior to care in an ambulance or ED (Table 7). EHRs indicated that 6 of these patients were treated with nitroglycerin without specifying self-administration. Two patients reported during CHT that pain relief occurred more than 15 min after self-administration of nitroglycerin. These patients are included in Table 7 in the set without complete relief of pain.

## Dimensionality of EHR and CHT datasets

We quantified the dimensionality of EHR data by searching EHRs for those with entries for a site of primary pain, a positive or negative finding for radiation to jaw or neck, a positive or negative finding for onset during physical activity, and a positive or negative finding for use of nitroglycerin. We included the criteria of a positive or negative finding because our data shows that missing EHR data may not reflect omission of negative findings. We found only 3 of 410 EHRs with explicit entries for the 4 attributes searched. Reducing the dimensions searched to any set of 3 of the above 4 expanded the number of complete EHRs only from 3 to 16. Four hundred nine CHT datasets included a positive or negative for all 4 attributes in the dimensionality analysis; 90% of CHT datasets had explicit positive or negative findings for all the data elements analyzed in all Tables and Figures in the preceding sections; and missing CHT data was related predictably to the question at which a CHT session was aborted.

## Discussion

We made a patient-by-patient comparison of EHR and CHT files for a set of data elements significant for the differential diagnosis of chest pain that are available only by history-taking from affected patients. We found, as compared with CHT files, that significant amounts of data were missing in EHRs; missing EHR data was not limited to absence of negative findings; and missing EHR data was distributed randomly across all EHRs and all data elements, which limited severely the dimensionality of EHR data. We also found frequent discrepancies of fact between EHR and CHT data for specific positive and negative findings. These findings are not unique to the environment in which our study was conducted or to the clinical problem patients presented; for histories in paper and EHR charts are known generally to have the same deficiencies as our analysis found [6–18]. There is, however, an important difference between the present and most prior analyses of data quality in physicians' histories. Thus, our work shows that CHT can address these issues in the context of a specific clinical problem.

## The advantages of CHT for history-taking are general

CHT has multiple, general advantages for history-taking as compared with physicians (Table 8). Computers running a CHT program have unlimited cognitive capacity and short-term memory [30]. Computers interacting with patients are not time-constrained and will not limit patient response times to seconds [16]. Expert system software is not programmed to use heuristics to collect and process history data [31, 32]. And, as compared with knowledge- and time-constrained physicians at the "bedside," expert system software for CHT is developed by panels of clinical experts working deliberatively, without time constraints, using System 2 thinking exclusively [33], and with immediate access to each other and the literature. The clinical experts formalizing their medical knowledge in the form of an expert CHT program also can test and edit their work because CHT leaves an exact record of how data was collected and what might have been missed in any interview. It appears, therefore, that the advantages of CHT technology for acquiring history data directly from patients can be adapted to address a large range of clinical problems [9, 10, 24–26].

Indeed, review of the CHT data in the current work identified an error in the CHT knowledge base, *e.g.*, patients with pain restricted to the posterior chest were not asked about radiation. EHR records are too noisy to allow detection of such discrete errors (6–21), not to mention the near impossibility of detecting and correcting such errors in the midst of history-taking by physicians. Still further advantages of CHT are standardization of history-taking by a single controlled protocol, leveraging of expert knowledge, and scaling of resources. CHT technology also is a mechanism for interviewing patients in their preferred languages while presenting findings and decision support in physicians' preferred languages. Of course, the value of CHT will have to be established on a problem-by-problem basis across the scope of medical practice. It will have to be shown too that CHT is acceptable to patients and that the reporting of CHT findings is acceptable to physicians. Considerably more academic, clinical research appears needed therefore for developing and testing CHT software.

## Limitations of the present study

**Data are from a single institution.** EHR data was extracted from records at a single hospital. Our results for physician histories may not be general; and the enrolled patients may not represent patients outside the capture area for Danderyd University Hospital.

**Data were collected only in the ED and addressed a single clinical problem.** Data in this study was collected exclusively in a busy ED. There is no certainty that the inadequacy of

**Table 8. Comparison of physician and CHT attributes affecting the completeness, accuracy and bias of data collected by each method for history-taking.**

| Attribute | Physician | CHT |
|---|---|---|
| Limited short term memory | Yes | No |
| Non-expandable cognitive capacity for learning impacted negatively by time-pressure and distraction | Yes | No |
| Use heuristics to collect and interpret data | Yes | No |
| Limited time with patient | Yes | No |
| Limit patient's answer-times to seconds | Yes | No |
| Leverage expertise and scale resources to need | No | Yes |
| Complete control of what is asked and what is not asked to standardize data collection | No | Yes |
| Automated data-entry into a structured, electronic database | No | Yes |
| History-taking in the patient's preferred language | Maybe | Always |

EHR data in an ED apply equally to ambulatory care. There is no certainty that the differences found between EHR and CHT data will apply to all other clinical problems.

**Incomplete CHT sessions.**   Relevant data was missing from some CHT files because some patients were discharged to home or admitted to hospital before completing CHT sessions. Also, some patients ended interviews prematurely at their discretion. No data was collected to determine factors influencing patients' decisions to enroll, continue with or discontinue a CHT interview. These issues need targeted study.

## Supporting information

**S1 Fig. Sites of primary pain recorded in EHRs and reported by patients during CHT.** Each line are sites of pain for the same patient. Column numbers refer to regions of the chest displayed in Fig 1. Blue-colored upper triangles for each location of pain are data from the patient's EHR. Blue-colored lower triangles for each location of pain are data from the patient's CHT interview. Empty upper triangles indicate that the region was not mentioned as affected by pain in EHR data. Empty lower triangles indicate that the region was not selected by the patient interacting with the image in S1 Fig. Absence of EHR data for location of chest pain indicates that the EHR did not record a specific site of chest pain. Blue-colored upper triangles in Column 0 reflect that the description of chest pain in EHR narratives was too imprecise to be associated with a specific anatomic region of the chest. Projected areas of pain on each line are data for the same patient.
(PDF)

**S2 Fig.**  Regions of primary pain projected onto an image of the chest for selected patients recorded in EHR data (left hand panels) and CHT data (right hand panels).
(PDF)

## Acknowledgments

The corresponding author attests that all listed authors meet authorship criteria and that no others meeting the criteria have been omitted.

## Author Contributions

**Conceptualization:** David Zakim, Thomas Kahan, Carl Johan Sundberg.

**Data curation:** David Zakim, Sami El Amrani, Andreas Hultgren, Natalia Stathakarou, Sokratis Nifakos.

**Formal analysis:** David Zakim.

**Funding acquisition:** David Zakim, Thomas Kahan, Carl Johan Sundberg.

**Investigation:** David Zakim.

**Methodology:** David Zakim, Carl Johan Sundberg.

**Project administration:** David Zakim.

**Resources:** David Zakim, Sokratis Nifakos, Carl Johan Sundberg.

**Software:** David Zakim, Sabine Koch.

**Supervision:** David Zakim, Helge Brandberg, Thomas Kahan, Jonas Spaak, Carl Johan Sundberg.

**Writing – original draft:** David Zakim.

**Writing – review & editing:** David Zakim, Helge Brandberg, Sami El Amrani, Andreas Hultgren, Natalia Stathakarou, Sokratis Nifakos, Thomas Kahan, Jonas Spaak, Sabine Koch, Carl Johan Sundberg.

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
