## [Decision Letter · Decision Letter 0]

27 Apr 2021

PONE-D-21-11324

Computerized History-Taking Improves Data Quality for Clinical Decision-Making. Comparison of EHR and Computer-Acquired History Data in Patients with Chest Pain.

PLOS ONE

Dear Dr. Zakim,

Thank you for submitting your manuscript to PLOS ONE. After careful consideration, we feel that it has merit but does not fully meet PLOS ONE’s publication criteria as it currently stands. Therefore, we invite you to submit a revised version of the manuscript that addresses the points raised during the review process.

We look forward to receiving your revised manuscript.

Kind regards,

Amit Bahl

Academic Editor

PLOS ONE

2. In your Methods section, please provide additional information about the participant recruitment method and the demographic details of your participants. If materials, methods, and protocols are well established, authors may cite articles where those protocols are described in detail, but the submission should include sufficient information to be understood independent of these references (https://journals.plos.org/plosone/s/submission-guidelines#loc-materials-and-methods).

Thus, please ensure you have provided sufficient details to replicate the analyses such as:

a) the recruitment date range (month and year),

b) a description of any inclusion/exclusion criteria that were applied to participant recruitment,

c) a table of relevant demographic details,

d) a statement as to whether your sample can be considered representative of a larger population,

e) a description of how participants were recruited, and

f) descriptions of where participants were recruited and where the research took place. Moreover, please ensure that the tool used has been described in sufficient detail.

4. We note that you have a patent relating to material pertinent to this article.

a. Please provide an amended statement of Competing Interests to declare this patent (with details including name and number), along with any other relevant declarations relating to employment, consultancy, patents, products in development or modified products etc.

Please confirm that this does not alter your adherence to all PLOS ONE policies on sharing data and materials, as detailed online in our guide for authors http://journals.plos.org/plosone/s/competing-interests by including the following statement: "This does not alter our adherence to  PLOS ONE policies on sharing data and materials.” If there are restrictions on sharing of data and/or materials, please state these.

Please note that we cannot proceed with consideration of your article until this information has been declared.

Reviewers' comments:

Reviewer's Responses to Questions

**Comments to the Author**

1. Is the manuscript technically sound, and do the data support the conclusions?

Reviewer #1: Partly

Reviewer #2: No

Reviewer #3: Partly

2. Has the statistical analysis been performed appropriately and rigorously? 

Reviewer #1: No

Reviewer #2: No

Reviewer #3: No

3. Have the authors made all data underlying the findings in their manuscript fully available?

Reviewer #1: Yes

Reviewer #2: No

Reviewer #3: Yes

4. Is the manuscript presented in an intelligible fashion and written in standard English?

Reviewer #1: Yes

Reviewer #2: Yes

Reviewer #3: Yes

5. Review Comments to the Author

Reviewer #1: The authors conducted a prospective observational study where they compared the data collected by physicians during an ED encounter for chest pain vs the data collected by a computerized history taking software. They enrolled 410 patients and concluded that computerized data was more complete and more representative of patient perceptions. They also suggest that computer generated histories will be more beneficial for aggregated data bases to be used in research. The authors continue to mention that physician histories are inaccurate and incomplete to an extent that clinical outcomes and patient care are effected by this. However, they present no data here to suggest that this is the case. Overall, the manuscript is well written. However, there are issues with broad overarching conclusions based off of likely clinically irrelevant differences between CHT and EMR data. This study is very relevant to EMR based population studies and shows how CHT can significantly increase the value of research using this methodology. If the language and conclusions are paired down to only include what was evaluated within this study, then this manuscript should be re-considered.

Points

1. In methods, need to describe the clinical setting and selection criteria rather than referencing previous work without any description.

2. Need to include the explicit question of pain that was shown in CHT. Looking at figure 1 this is quite different then what is described in the body of your paper.

3. Multiple mentions of improving patient outcomes. Authors seem to suggest that CHT, given more available data, will lead to better patient care. However, there is no room for these conclusions to be made from this paper. They have no data on accuracy of diagnosis, patient outcomes, or patient satisfaction. Therefore it seems premature to make any comment on CHT improving patient care.

4. Further analysis of the data from the 20 interrupted CHT sessions should be examined for level of agreement between EMR and CHT. It would be interesting to see if after talking to a physician there was more agreement between the 2 history taking methods. It is possible that the physician could clarify points that the CHT was unable too leading to differences in CHT answers after interview.

5. If you are going to excluded data from 2 patients who appeared to make deliberately false entries during CHT, this data should be made available to the reader so that they can understand your conclusion.

6. The description of how patient’s selected CP location that is currently in the results section and should be moved into methods. Additionally, the description currently seems overly complicated and verbose.

7. A major sticking point of the authors is specific location of chest pain. They note that EHR data was documented as left, right, or central but rarely specified upper or lower. The main issue I take here with calling this “incomplete” EHR data is that when given more options to specify upper or lower, the patient likely will, as we see with the CHT data. However, the clinical relevance to such specificity is likely little to none. Looking at figure 2 the CHT system of location classification is extremely over complicated from a clinical perspective. I would recommend re-evaluating your data based on more reasonable expectations from a clinical standpoint. Consider dividing the locations into L and R anterior chest, L and R arm, and back. 15 locations to describe chest/torso pain would never be something reasonable or clinically relevant to document from a physicians perspective. Another option would to be rather than comparing a clinically meaningless variable to a clinical one (EHR hx), just present how much more granular data given in CHT can be. In this way you could make the argument that this level of granularity may at some point show clinical relevance and therefore must be studied. Currently I do not know of any literature that would suggest this level of granularity is of any clinical value.

8. This theme is repeated in the discussion regarding pain radiation to the arm. Authors note that EHR indicated isolated radiation to the arm, while CHT indicated arm + shoulder. This information again seems clinically irrelevant and likely highlights the point that CHT was collecting unnecessary data that would have slowed down physician history taking. Additionally, even if mentioned by the patient, this likely something that would have been documented in a truncated fashion by the physician in order to save time during EMR note composition. If the goal was to demonstrate fallibility of the EMR physician history, they you would need to specifically analyze what was said by the patient and then what was documented by the physician. In clinical encounters it is not uncommon to exclude portions of the history that you feel are irrelevant or combine portions during authoring of your HPI in order to save time.

9. Authors note that CLEOS was programmed to ask about radiation only when patient had anterior chest pain therefore 6 patient’s did not have radiation data in this group. This concept perfectly illustrates the issue with concluding that CHT is more complete and beneficial to patient care than physician history taking. Many experienced clinicians have the same “programming” when it comes to history taking and only include or exclude questions based on overall gestalt and patient complaint. When presented with a limited amount of time for history taking the physician must rely on their ability to expand or contract their history taking in order to ask the most relevant and pertinent questions.

10. Sup Figure 1 and Figure 3 are very complicated and difficult to understand easily. Since this is a key component of your argument would recommend rethinking this figure.

11. Figures from CHT are poor resolution making them difficult to read.

12. Overall, the strongest point of this paper is that EHR data is likely inadequate for any population-based studies and that CHT could improve this process. It is overreaching and likely inaccurate that CHT improves any sort of patient outcomes or that physician history taking is deficient. There is certainly no data within this manuscript to suggest that CHT may have lead to any improvement in quality of patient care.

Typos: Page 5 “The problem was corrected during the course of this stud.”

Reviewer #2: The authors have developed a tool for patients to use to enter their clinical history electronically, when they attend an emergency department with chest pain. They have tested this by enrolling patients at a single site, university hospital in Sweden, who attended their ED with chest pain, who didn't have a diagnostic first ECG on arrival. Patients were asked to enter clinical data into the tool, which was compared to the medical notes made by their attending physician after they had seen the patient. They enrolled a convenience sample of 410 patients. They then compared the patient authored data to the physician medical notes in the Electronic Health Record. They conclude that there is a lot of variation between the patient and physician notes, then state that their tool is better than the physician evaluation of the patient.

Major issues:

The manuscript has multiple major issues. Some are domain issues, others relate to the write-up.

1. The structure of the manuscript - I strongly suggest that the authors review the appropriate equator guideline and rewrite their manuscript accordingly. This is a comparative observational study, so STROBE would be appropriate. If this is done, the authors will realise that multiple mandatory areas of information are absent from each section. They should end with re-writing the abstract into a structured format. Given that there is machine learning involved in the development of the tool, the authors should also consider reporting elements of the TRIPOD guidelines.

2. There seems to be a major misunderstanding by the author group about the role of medical notes. This has made their work fundamentally flawed, despite the valid work that has gone into their research. Medical notes are not a verbatim record of what the patient said, nor are they usually intended to be a complete record - physicians don't have time and would record medical interviews if this was required. Medical notes are a synthesis or interpretation of information provided from various sources, aiming at being concise and useful in explaining ongoing medical decision making. Physicians deliberately exclude some history features from their notes, when the patient thinks they have a symptom, but when clarified, the symptom is different to the one sought by the physician - for example pleuritic chest pain. Comparing patient reported symptoms to a medical note is not a reasonable comparison. Whilst it is very possible that physicians have missed important aspects of the history during their consultations, this hasn't been proven by this research.

The authors might be better off asking a different research question: If the physician is presented with the patient reported information to read prior to/during/after the consultation, does this change the medical management or decision making or differential diagnosis at all? It is possible that it does change management and that physicians don't always ask enough detail about patient symptoms, but this hasn't been investigated so far and would be much more important to physicians.

3. The length of the manuscript and the discussion is overwhelming. A more concise version may be better received by readers. In contrast, the limitations section is too short, seemingly failing to understand the limitations of the study.

4. If readers are to understand the research, some illustrative examples should be provided of the typical outputs obtained by the tool, compared to the medical notes.

Suggested write up improvements:

Abstract: Many important details are missing from the abstract. This includes a description of the aims, population, setting, intervention (and that you developed it and own the patent), demographics of participants, methods of comparison, and so on. You seem to have simply written up your conclusions. You have no evidence that your tool outperformed clinicians, simply that the data you compared were different.

Introduction: A well structured, medical manuscript introduction should be about 4 paragraphs, broken up into the initial description of the problem and its magnitude; what is currently known; what is the gap in the literature and why does it matter and then a short goals of the investigation section.

Methods:

Each section needs improvement (or creation). It is not enough to state that you have published a study protocol, and not to describe your setting and population. Readers shouldn't have to look up another study for such basic information. You need a statistical plan - sample size calculation, statistical methods etc. Please refer to the equator guidelines +/- TRIPOD for more information

Results:

This should start with a flow diagram or statement, beginning with how many patients attended your ED with chest problem (either at triage or as a discharge diagnosis), how many patients were screened, how many were excluded and why, how many were enrolled, how many complete datasets (per participant) you obtained, how many were analysed, how many weren't and why etc. (see CONSORT guidelines for examples)

Next you require a section on your participant demographics, compared to those you didn't enrol.

Finally the main results should be presented. I would move much of your description to an appendix, the length is overwhelming to read. I would suggest that the narrative text that remains is more concise and less judgemental - present that something was present or absent from one set of notes or another. You have some odd groupings to your section sub-headings. Why for example is there a heading about both sweating and pleuritic pain, this should be broken up as the symptoms are completely unrelated clinically.

Discussion

Much of the discussion reads as though there is a perception that the patient notes are better (more accurate) and more useful. At best the authors can state that the data was different. Heuristics are extremely important in clinical medicine, to effectively write this off is deeply flawed. The medical evaluation and the medical notes are intended to be a synthesis and the way the manuscript is written seems to have completely misunderstood this point.

Much of the discussion is giving unproven opinions. I would suggest that as this manuscript is intended for a medical audience (I assume) that the structure is 1. a short summary of important results 2. A comparison to previous literature (does it confirm or refute previous work) 3. external generalisability (or lack of same) 4. limitations and future work

It would be important to discuss multiple limitations to this work in the discussion. One very important point is that patients rarely present to the ED with differentiated, well defined symptoms - such as chest pain. Many patients, even those with chest pain, wouldn't have only chest pain. To only obtain a history regarding their chest symptoms, would be a mistake that will lead to medical errors.

Reviewer #3: The manuscipt addresses an important question in this era of computerized medicine - does the traditional EHR record compare favorably to a patient-entered computerized history taking program. However the manuscript in the current form has several critical issues that must be addressed in order to be publishable.

Abstract:

The abstract has no background or context for the study. It dives right into methods. What do we know about computerized history taking? Why is it important to compare computerized history taking to the standard EHR history? What were the primary and secondary aims of the study?

Introduction:

This needs some major editing. There are many unclear statements such as: "Moreover, objective data cannot yet supplant medical history data". The introduction includes results and interpretation which are inappropriate for this part of the manuscript. This needs to be restructured: what is the background, what is unknown about the problem, what knowledge holes are you hoping to fill in, what are your primary and secondary aims?

Materials and Methods:

Also needs major edits. Clinical setting, selection criteria, and recruitment need to be spelled out here, not listed in a separte document. There needs to be an explanation of how the EHR history was taken by the physician and entered into the computer as well. There needs to be a discussion of the statistical analysis. Finally, the last sentence (Four patients did not...) should be in the results section.

Results:

This is far too long. Please summarize the major findings and point to the specifics in your tables. Also, the tables need improvement. What are the Ns for each grouping? Are the congruences reported statistically significant? Also there is occasionally author interpretation in these results which should be in the discussion

Discussion:

This discussion veers way off topic and needs improvement. This isn't a review article about the challenges of history taking. This is a study comparing EHR vs computerized history taking and the discussion should reflect that. How do these findings fit in with what is known? What remains to be discovered? Is this clinically relevent? Maybe physicians obtain what they need to in order to make the correct diagnosis and the rest is not needed. Maybe the physicians do ask the questions but don't waste their time on entering it into the EHR. The statement "It is fair to argue that history-taking by physicians has become an inefficient use of the physician's time with the patient" is not supported - what about building rapport? What about the nuance that comes from non-verbal or non-written conversation?

Finally:

There are multiple spelling and typographical errors that need to be addressed.

6. PLOS authors have the option to publish the peer review history of their article (what does this mean?). If published, this will include your full peer review and any attached files.

Reviewer #1: No

Reviewer #2: No

Reviewer #3: No

---

## [Author Response · Author response to Decision Letter 0]

19 May 2021

Reply to Reviewer #1. Reviewer’s words are in Italics.

Summary statement.

a. The reviewer concludes that we describe results from an observational study. This is correct only in the sense that all experiments involve "observations." An "observational study" in the clinical context refers, however, to a study in which clinical outcomes are measured in the absence of matched experimental and control cohorts. Our work is a "within person" comparison of CHT and EHR chest pain data. Outcomes for enrolled patients were not captured. We clarify the design of the study in the revised manuscript by adding an explicit statement about the study design as the first paragraph of the Methods section. 

b. The reviewer states: The authors continue to mention that physician histories are inaccurate and incomplete to an extent that clinical outcomes and patient care are effected[sic]by this. 

This comment misstates what the text includes and ignores the evidence cited to support this text. We do not aver but cite the evidence in the Introduction that medical history data is the salient data for diagnostic decisions, that history-taking by physicians is inadequate and that clinical errors are frequent. We assumed that the connection between these factually correct statements i.e., citations of the evidence in the literature, is obvious to the biomedical community and did not explicitly connect poor history-taking with clinical errors in the original text. We do so in the edited text and cite the literature supporting this connection.

c. We did not elaborate in the original text that poor history-taking, including the falsification of findings, is documented by post hoc review of records physicians generate and by real-time observation of physicians interviewing patients. This work was cited in the original text but not mentioned separately. The edited text elaborates that there are two sets of observations documenting poor history-taking by physicians: (i) post hoc analysis of medical records and (ii) real-time observation of physicians interviewing patients. 

d. The reviewer states: ... there are issues with broad overarching conclusions based off of likely clinically irrelevant differences between CHT and EMR data. 

The original text compared EHR and CHT files for a range of data elements that are relevant now to the differential diagnosis and triage of patients with chest pain. Our original text also described the full range of differences between EHR data and CHT data for precise locations of primary and radiated chest pain. The reviewer seems only to raise a question about differences between EHR and CHT data for elements that appear to have no clinical relevance today. We have addressed this criticism in the following way. i. . Results for primary and radiated pain in the edited text are restricted, nearly exclusively, to data elements relevant to this differential and to triage algorithms for patients with chest pain. ii. We have removed almost all text for primary and radiated pain that does not include areas and patterns of pain applicable to guideline-based management of chest pain. iii. We have removed one table and one figure to focus the data for locations of pain on guideline-based patterns. iv. We mention only in passing that CHT data is more precise as compared with EHR data and refer the reader to Supplementary Fig 1.

Numbered items.

In methods, need to describe the clinical setting and selection criteria rather than referencing previous work without any description.

Response: These details have been added to Methods and not simply referenced. 

2. Need to include the explicit question of pain that was shown in CHT. Looking at figure 1 this is quite different then what is described in the body of your paper.

Response: We do not understand the problem the reviewer mentions. It seems the reviewer might have confused Figs. 1 and 2. We have reviewed the text citing these Figs and their legends. Their meaning seems clear.

3. Multiple mentions of improving patient outcomes. Authors seem to suggest that CHT, given more available data, will lead to better patient care. However, there is no room for these conclusions to be made from this paper. They have no data on accuracy of diagnosis, patient outcomes, or patient satisfaction. Therefore it seems premature to make any comment on CHT improving patient care.

 Response: The word "outcomes" appears only once in the original text (in the Introduction). We have removed the word "outcomes" from the edited Introduction. The word "outcomes" appears otherwise in the titles of 4 journal articles in the original reference list. These references were to studies of the effectiveness of EHR-based automated clinic al decision support. The edits to the text no longer refer to any of these citations, which are absent from the edited reference list.

Moreover, we did not in the original and do not in the revised text state or suggest that CHT will improve diagnostic accuracy or outcomes. 

However, on the basis of substantial literature supporting it, we make this argument in the original and edited Introduction: Medical history data is essential for correct diagnostic and treatment decisions. Poor medical history data is associated with diagnostic errors. Thus, we need to develop better methods for history-taking. 4. 

4. Further analysis of the data from the 20 interrupted CHT sessions…

Response: We have added the relevant details to Methods in the revised manuscript. These details, i.e., discrepancies between EHR and CHT data for 20 records with a long interruption during CHT as frequent as in 390 other files, belong in Methods because they explain why data for all 410 patients was combined. 

5. If you are going to excluded [sic]data from 2 patients …

Response: We have added to Methods the specific data elements in the interviews of two patients that led us to exclude their data, which appeared to be corrupted deliberately. 

6. The description of how patient’s selected CP location that is currently in the results section and should be moved into methods. Additionally, the description currently seems overly complicated and verbose.

Response: We believe the manuscript reads more clearly with the Fig and its description in Results. We decided not to move Fig 2 and the description of how patients indicated areas of primary pain to Methods. 

We note, however, that DZ misread the author instructions for placement of figure legends and tables in the main text. He placed these immediately after the first mention of a figure or table and not, as instructed, in the first paragraph after the first mention. DZ's error undoubtedly contributed to the reviewer's interpretations of the text and legend for Fig 2 as ... overly complicated and verbose. 

DZ's errors in placing figure legends and Tables in the text are corrected in the revised manuscript. We did not edit the text referring to Fig 2, which is a simple sentence. "Patients reported pain during CHT using a version of Fig 2 without numbered anatomic areas." We did not edit the legend for Fig 2, which seems clear. 

7 and 8. A major sticking point of the authors is specific location of chest pain.

Response: Our response to the criticism that sites of pain were not clinically relevant and the edits made to the text are given in detail in our response to Summary statement, paragraph d. The fact of the matter is the original text described important differences between EHR and CHT data for sites of primary and radiated pain that are relevant to managing chest pain. As mentioned above, however, we have removed detailed comparisons between EHR and CHT data for sites of pain that are not known to be clinically relevant.

This theme is repeated in the discussion regarding pain radiation to the arm.

Looking at figure 2 the CHT system of location classification is extremely over complicated from a clinical perspective. 

This is true if the data has to be communicated orally to a physician, who has to remember the details of complex pain locations. It is not true in a CHT system. 

Our results show in fact that the physician probably does not collect and or does not recall simple pain locations accurately, e.g., 50% of EHRs did not cite a location for primary chest pain. Half the EHRs did not have an entry for radiated pain. The half without a site for primary pain was not the half without an entry for radiation. These points are emphasized in the edited text that compares EHR and CHT data, as regards primary and radiated pain, for presence/absence of central chest pain and citation of bilateral pain radiation or radiation to the right side or no radiated pain. 

In clinical encounters it is not uncommon to exclude portions of the history that you feel are irrelevant or combine portions during authoring of your HPI in order to save time.

This criticism of our view of the value of an accurate clinical record is precisely why we need programs like CHT to collect and record medical histories. 

For as we cite in the original and make more emphatic in the edited text, failure to enter negative findings does not explain the high frequency of the absence of clinically relevant positive findings in EHRs and does not account for the apparently false negative and false positive findings in the EHRs examined, e.g., no entry for site of pain, incorrect entries for which patients did or did not have central chest pain, false positive and false negative findings for pain beginning during physical activity, and so on.

DZ, TK, JS, HB and CJS are experienced clinicians. We have all saved time by shaving details from our clinical notes. But we acknowledge that electronic medical records are intended to serve as a mechanism to generate patient-specific clinical decision support and as databases for clinical research and that these uses imply the expectation that physicians will record all relevant data in patients' EHRs.

We all know, however, that we don't and in fact cannot meet this standard. So we need a method like CHT that accurately collects and records all relevant positive and negative findings.

9. Authors note that CLEOS was programmed to ask about radiation only when patient had anterior chest pain therefore 6 patients did not have radiation data in this group. This perfectly illustrates the issue with concluding that CHT is more complete and beneficial to patient care than physician history taking. 

Many experienced clinicians have the same “programming” when it comes to history taking and only include or exclude questions based on overall gestalt and patient complaint.

Response: The reviewer missed an obvious error in the programming of the CLEOS knowledge base. Patients with primary posterior chest pain and no primary anterior pain should be asked about radiation from back to front around the rib cage. The edited text mentions this error. We also have added a brief mention in the edited Discussion of the value of editing CHT software on the basis of clinical trials. 

We believe too that the reviewer’s opinion that … experienced clinicians have the same “programming”… is further affirmation of the value of CHT. Thus, even experienced physicians (the authors included) make errors of omission during history taking because “programming” the human mind does not insure failure- proof performance. By contrast, CHT may make errors; but once a program error is corrected, CHT will not make the same error again. 

10. Supplemantary [sic]Figure 1 and Figure 3 are very complicated and difficult to understand easily. Since this is a key component of your argument would recommend rethinking this figure.

Response: Supplementary Fig 1 has > 15,000 data fields for locations of primary pain for 410 patients in EHR and CHT records. We see no simpler way to display this amount of data than the scheme of Supplementary Fig 1. Moreover, Supplementary Fig 1 is not ... a key component of [our] argument.

The data in this figure were extracted and converted to text and Tables in the original and edited versions. The reader thus does not need to use the figure to understand the presentation of results for primary pain in the central chest. The interested reader, however, can use the figure to verify our findings. Moreover, the figure shows in a simple way, by comparison of upper and lower colored triangles across each line, the "granularity" of CHT as compared with EHR data. It also shows by inspection the extent and specifics of what is missing in EHR data as compared with CHT data.

We have removed the original Fig 3, which displays a portion of the data in Supplementary Fig 1.

11. Figures from CHT are poor resolution making them difficult to read.

Response: The reviewer is correct. We submit a better version of this Figure with the edited version. 

12. There is certainly no data within this manuscript to suggest that CHT may have lead[sic] to any improvement in quality of patient care.

Response: We agree that there is no such data in our manuscript. And whereas this comment implies we claim CHT will improve outcomes, the original text makes no such claims. We only cite, in the Introduction, the literature reporting that medical history remains the salient data for diagnosis, that physician-acquired histories are incomplete and generally inaccurate and that poor history taking is associated with diagnostic error. We then state that these undisputed findings point to the need for better history-taking. We state in the Discussion that the data analyzed indicates that the 410 EHRs reviewed are inadequate to support meaningful decision support or to be a resource for future clinical research. We also state that the CHT data would support these functions. The data analyzed align with these conclusions. And as mentioned in our response to the reviewer’s item 3, we draw no conclusions about the effect of CHT on patient outcomes. 

Reply to Reviewer #2. Reviewer’s words are in Italics.

Summary statement.

a. The authors have developed a tool for patients to use to enter their clinical history electronically... 

Patients were asked to enter clinical data into the tool...

They conclude that there is a lot of variation between the patient and physician notes,

We believe the reviewer either did not understand the experiment we report or was unable to explain his understanding articulately. Patients did not simply enter their clinical history electronically... We collected no patient ... notes. No patient authored data was collected.

We might have contributed, however, to the reviewer's possible misunderstanding of CLEOS and expert system software generally by referencing details of the CLEOS program that we should have described more fully in Methods. 

We have added a fuller description of how CLEOS operates to interview the patient. 

Patients interacted with expert system software programmed to emulate the clinical reasoning of a knowledgeable clinical expert collecting a medical history from a patient with acute chest pain. The expert system software used the same pathophysiologic reasoning to pose the same types and range of questions that an expert physician would employ while interviewing the same patient. Thus data entered by the patient during CHT consists of structured data captured by systematic patient interview.

b. They enrolled a convenience sample of 410 patients.

We consecutively enrolled patients presenting with chest pain in the ED. This was not explicitly stated in the original text but is clarified in the revised text. We do not understand what the reviewer means by a convenience sample. 

c. They conclude that there is a lot of variation between the patient and physician notes, then state that their tool is better than the physician evaluation of the patient.

This statement is imprecise and incorrect.

(i). There were no patient notes in the data extracted. 

(ii). We do not conclude ... there is a lot of variation between the patient and physician notes. Our results show clearly that EHRs were missing for multiple, specific data elements that are significant for the differential diagnosis of chest pain and that this data was present in CHT files for corresponding patients. We also show that besides the absence of relevant data fields, EHRs had large numbers of false negative and false positive entries by comparison with the data collected by CHT interview for clinically relevant information known only to the patient. 

The reviewer's apparent confusion about what the text states, to which we refer in (i) and (ii), is addressed in the edited version by the fuller description in Methods of how CLEOS is designed and operates.

(iii) We do not state that their [our] tool is better than the physician evaluation of the patient. This comment does not reflect what the text states. We report only on a comparison of EHR and CHT medical history data. Our text is mute on physician evaluations of patients. The reviewer's criticism might reflect a jump in thinking in which the reviewer connected our findings to the value of history data in evaluating patients. 

Numbered items.

The structure of the manuscript - I strongly suggest that the authors review the appropriate equator guideline and rewrite their manuscript accordingly. This is a comparative observational study, so STROBE would be appropriate. If this is done, the authors will realise that multiple mandatory areas of information are absent from each section. They should end with re-writing the abstract into a structured format. Given that there is machine learning involved in the development of the tool, the authors should also consider reporting elements of the TRIPOD guidelines.

Response: There is no appropriate Equator guideline for the experimental design we used. 

The reviewer concludes that we describe results from an observational study. This is correct only in the sense that all experiments involve "observations." 

An "observational study" in the clinical context refers to a study in which clinical outcomes are measured in the absence of matched experimental and control cohorts. Our work is a "within person" study. In addition, we did not measure any clinical outcome. We clarify the design of the study in the revised manuscript by adding an explicit statement about the study design as the first paragraph of the revised Methods section. However, we have reorganized the text of the Abstract, Introduction, Methods and Discussion sections along the Equator lines. 

Given that there is machine learning involved in the development of the tool...

Machine learning is not mentioned directly or by reference in the manuscript. Machine learning was not used to develop the expert system software used for CHT. CLEOS does not include any machine learning algorithms. These issues are clarified in the edited Methods section, which emphasizes that CLEOS interviews strictly according to pathophysiologic concepts.

2. There seems to be a major misunderstanding by the author group about the role of medical notes.

DZ, TK, JS, HB and CJS are experienced clinicians. We have all saved time by shaving details from our clinical notes. We acknowledge that electronic medical records are intended to serve as a mechanism to generate patient-specific clinical decision support and as databases for clinical research. These uses imply the expectation that physicians will record all relevant data in patients' EHRs.

We all know they don't and in fact cannot meet this standard. So we need a method like CHT that accurately records all relevant positive and negative findings. The criticism is the same as item 8 in review #1. Our reply to reviewer #2 on this issue is the same as our reply to reviewer #1.

Physicians deliberately exclude some history features from their notes, when the patient thinks they have a symptom, but when clarified, the symptom is different to the one sought by the physician...

The reviewer is correct on this point. But the idea expressed here - that physicians can divine what patient's experience - is precisely why we need CHT to collect and record medical histories. Indeed, we cited literature in the original text documenting that physicians purposely report findings that are contrary to factual information reported by their patients (c.f., Mamykina L et al J Am Med Inform Assoc 2012; 19:1025–31; Berdahl CT et al. JAMA Network Open. 2019; 2(9):e1911390; Farmer SA, et al. Ann Emerg Med 2006; 48: 78-85). These cites were in the original manuscript but were combined with the literature for post hoc analysis of clinical records. We have edited the textual reference to these papers to highlight that they deal with real-time observations of physicians corrupting data as they collect it. 

Comparing patient reported symptoms to a medical note is not a reasonable comparison.

We did not compare “notes.” We compared entries for specific data elements in EHRs with entries for the same specific data elements in CHT files for data elements that can be reported only by affected patients, e.g., the site of their chest pain. Methods in the original and revised versions state explicitly how the data analyzed was extracted from EHR and CHT datasets. 

3. The length of the manuscript and the discussion is [sic] overwhelming. A more concise version may be better received by readers. In contrast, the limitations section is too short, seemingly failing to understand the limitations of the study

Response: The reviewer's suggestions for revising the manuscript are replete with misstatements of fact about the original manuscript.

We have changed the heading that confused the reviewer to Associated Symptoms.

The authors might be better off asking a different research question: If the physician is presented with the patient reported information to read prior to/during/after the consultation, does this change the medical management or decision making or differential diagnosis at all?

Obviously, this would be a significant experiment. However, it not the first experiment to run in developing expert systems for CHT. 

4. If readers are to understand the research, some illustrative examples should be provided of the typical outputs obtained by the tool, compared to the medical notes.

Response: The text describes how patients entered data for sites of primary pain by interacting with Fig 2. 

We have added a description to Methods on the development of questions and answers, how patients select answers and how their entries are saved. In brief, the data fields analyzed were simple linguistically. For example, location of primary pain; onset during physical exercise yes/no; self-administered nitroglycerin yes/no. Methods explain how the data was acquired and how it was extracted from EHR and CHT data. The contents of Results are the typical outputs that were extracted from CHT and EHR records for specific data elements. 

Suggested write up improvements:

Abstract

We have added the information to the abstract that reviewer #2 suggests we add.

Introduction

We believe the original text met the reviewer's suggestions. But we have shortened the edited Introduction. 

Methods 

Also needs major edits. Clinical setting, selection criteria, and recruitment need to be spelled out here, not listed in a separte [sic] document. There needs to be an explanation of how the EHR history was taken by the physician and entered into the computer as well. There needs to be a discussion of the statistical analysis. Finally, the last sentence (Four patients did not...) should be in the results section.

As noted above, we have included in the edited methods section sufficient information to replicate the experiment we report without reference to other literature. 

There needs to be an explanation of how the EHR history was taken by the physician and entered into the computer as well. 

We have added the explicit statement that physicians interviewed patients while delivering routine care. 

Finally, the last sentence (Four patients did not...) should be in the results section.

The data for the 2 not 4 patients to whom the reviewer refers was not included in the analyses presented in Results. 

Results

i. This should start with a flow diagram or statement,beginning with how many patient...

These details belong in Methods, which has been edited as described above. 

ii. The length of the text is overwhelming to read.

The original text had no fluff. The edited text is shorter because we have removed details that are not known at present to be clinically relevant to the differential of chest pain.

iii. I would suggest that the narrative text that remains is more concise and less judgemental [sic] - present that something was present or absent from one set of notes or another.

The text in Results, original and edited versions, contains statements of fact in regard to the data examined, i.e., data was missing in EHRs, data in EHRs was inconsistent with the same data in the same patient's CHT interview. 

Discussion

i. Much of the discussion reads as though there is a perception that the patient notes are better (more accurate) and more useful.

There are no patient notes.

That said, we review a set of data elements that are patients' perceptions of the location of primary pain, the location of radiated pain, the setting in which their pain began, when it began, whether it was episodic or constant, and so on. We compare these elements as recorded in EHRs and as recorded automatically as the elements were collected by interactions between the patient and an expert system, as described above. We found that most of the data reviewed was missing from EHRs and that much of the data that was present in EHRs was contrary to patients' perceptions of their pain that were collected by CHT. Of course, the discussion of these findings, in the context of an abundance of literature indicating that physician histories are incomplete, inaccurate and sometimes deliberately will convey the perception that the patient knows more about what they feel than their physician does.

ii. Heuristics are extremely important in clinical medicine, to effectively write this off is deeply flawed.

The medical and psychological literature show clear, negative effects of heuristics on decision making inside and outside the sphere of clinical medicine. The edited version cites the literature explicitly on this issue. 

iii. Much of the discussion is giving unproven opinions.

This criticism is a non-sequitur. Opinions are not facts. So the criticism the reviewer wants to make is that the Discussion is mostly opinion. Most of the Discussion reviews the findings in the context of prior work on the quality of physician-acquired histories. We briefly speculate on the long-term significance of our results. We believe this is one of the purposes of a Discussion section, i.e., what might be the long-term significance of the work reported. 

iv. It would be important to discuss multiple limitations to this work in the discussion. One very important point is that patients rarely present to the ED with differentiated, well defined symptoms - such as chest pain.

The complexity of these problems highlights the need to collect highly detailed information from the patient. Since half the author group comprises physicians with clinical experience, the authors understand first-hand the complexity of the problems presented by sick patients. 

Reply to Reviewer #3. Reviewer’s words are in Italics.

Abstract

The abstract has no background or context for the study. It dives right into methods. What do we know about computerized history taking? Why is it important to compare computerized history taking to the standard EHR history? What were the primary and secondary aims of the study?

Response: We have edited the abstract to include the items the reviewer suggests and to make it less abrupt.

Introduction

Moreover, objective data cannot yet supplant medical history data.

This is a quote from the original manuscript. We have edited it out of the revised manuscript.

... What is the background, what is unknown about the problem, what knowledge holes are you hoping to fill in, what are your primary and secondary aims?

We have reread the Introduction and conclude it includes all the elements asked for here. However, we have edited the language of the Introduction to bring attention more effectively to these issues. 

The introduction includes results and interpretation which are inappropriate for this part of the manuscript.

Contrary to the reviewer's opinion, this is an acceptable style in the physical science and biomedical literature. DZ, the author of the first draft of the manuscript, has written Introductions to his papers in this style for 60 years. We retain this style in the edited version but have shortened the text.

Materials and Methods

Also needs major edits. Clinical setting, selection criteria, and recruitment need to be spelled out here, not listed in a separte [sic] document. There needs to be an explanation of how the EHR history was taken by the physician and entered into the computer as well. There needs to be a discussion of the statistical analysis. Finally, the last sentence (Four patients did not...) should be in the results section.

We have added all details to Methods that previously were referenced. We also have added clarifications about the nature of the CLEOS software program. These edits include reviewer #3's reference to 4 patients in the last line of Methods. The correct and cited number is two not four patients. We keep the text on excluding data for these two of 410 patients in Methods because their data was excluded from analysis (for the detailed reasons added to the edited version). 

Results

Results have been edited for clarity and to remove detailed analysis of data elements not immediately relevant to the differential of chest pain. 

Our Results section presents major findings. We summarize these briefly, as the reviewer suggests, in the Abstract and at the beginning of the Discussion. 

What are the Ns for each grouping?

The Tables and text clearly indicate the numbers of patients in each group and with each finding. 

Are the congruences [sic] reported statistically significant?

If there were statistical noise in answers patients provide when asked about their symptoms, e.g., the site of chest pain, then medical history data obtained by any method would not be useful for any clinical purpose. In addition, the original and edited texts state how we looked for evidence, that patient’s perceptions of the site of pain might have changed across short time intervals. We mentioned in the original manuscript that the relevant evidence indicated that patient-reported pain patterns were stable. 

The Reviewer's question about statistical significance called our attention to an error in our calculations of congruence. We calculated per cent congruence across the entire set of 410 patients. Congruence should have been calculated as the number of patients found by EHR data with a specific data element divided by the number of patients with the same specific data element found in CHT data. We have corrected accordingly the tables and text in the Revised manuscript.

... there is occasionally author interpretation in these results which should be in the discussion.

The Results section has text other than table and figure legends. But we do not understand what the reviewer refers to as "interpretation."

Discussion

How do these findings fit in with what is known?

This is covered in the original Introduction and Discussion. It is covered at this reviewer's suggestion as well in the edited Abstract of the edited paper.

Maybe physicians obtain what they need to in order to make the correct diagnosis and the rest is not needed. 

The manuscript cites extensive literature in the Introduction and again in the Discussion that shows there is no maybe about it. The literature establishes that physicians generally do not get the data that is needed! 

Our Results show specifically for selected data elements in a large sample of patients that physicians do not record complete or apparently accurate data that is immediately relevant to the differential diagnosis of acute chest pain.

Maybe the physicians do ask the questions but don't waste their time on entering it into the EHR.

As we cite in the original and more emphatically in the edited text, failure to enter negative findings does not explain the high frequency of discrepancies between EHR and CHT for clinically relevant, apparently false negative and false positive findings in the EHRs examined, e.g., no entry for site of pain, incorrect entries for which patients did or did not have central chest pain, false positive and false negative findings for pain beginning during physical activity, and so on. Failure to enter negative findings also does not explain the high frequency of missing EHR data for positive findings reported by patients during CHT. 

The Reviewer objects to our statement: "It is fair to argue that history-taking by physicians has become an inefficient use of the physician's time with the patient." 

We have edited out this statement. 

The reviewer asks ...what about building rapport? What about the nuance that comes from non-verbal or non-written conversation?

The reviewer seems to have the jump to thinking that CHT will replace physicians. We do not suggest or imply that CHT will replace physicians. The manuscript points out instead, however, that CHT can enhance the value of the physician-patient interaction for physician and patient and especially contribute to patient trust, which is important for patient compliance.

---

## [Decision Letter · Decision Letter 1]

24 Jun 2021

PONE-D-21-11324R1

Computerized History-Taking Improves Data Quality for Clinical Decision-Making. Comparison of EHR and Computer-Acquired History Data in Patients with Chest Pain.

PLOS ONE

Dear Dr. Zakim,

Thank you for submitting your manuscript to PLOS ONE. After careful consideration, we feel that it has merit but does not fully meet PLOS ONE’s publication criteria as it currently stands. Therefore, we invite you to submit a revised version of the manuscript that addresses the points raised during the review process.

Thank you for reviewing and responding to the reviewer concerns. However, additional modifications are needed. Please note reviewer comments. Id advise you to pay special attention to common themes among reviewers as this usually signifies an area of opportunity. Particularly, the results section needs to be pared down to be more organized and sensible. Some data may be better displayed in tables - I strongly recommend moving some data to supplementary or supporting tables/figures. I hope you will consider the comments and make adjustments accordingly. 

We look forward to receiving your revised manuscript.

Kind regards,

Amit Bahl

Academic Editor

PLOS ONE

Reviewers' comments:

Reviewer's Responses to Questions

**Comments to the Author**

1. If the authors have adequately addressed your comments raised in a previous round of review and you feel that this manuscript is now acceptable for publication, you may indicate that here to bypass the “Comments to the Author” section, enter your conflict of interest statement in the “Confidential to Editor” section, and submit your "Accept" recommendation.

Reviewer #1: All comments have been addressed

Reviewer #2: (No Response)

Reviewer #3: (No Response)

2. Is the manuscript technically sound, and do the data support the conclusions?

Reviewer #1: Yes

Reviewer #2: No

Reviewer #3: Partly

3. Has the statistical analysis been performed appropriately and rigorously? 

Reviewer #1: Yes

Reviewer #2: I Don't Know

Reviewer #3: No

4. Have the authors made all data underlying the findings in their manuscript fully available?

Reviewer #1: Yes

Reviewer #2: (No Response)

Reviewer #3: Yes

5. Is the manuscript presented in an intelligible fashion and written in standard English?

Reviewer #1: Yes

Reviewer #2: (No Response)

Reviewer #3: No

6. Review Comments to the Author

Reviewer #1: In my first review of this paper the main issue I took was with a broad over-reaching conclusion based off of likely clinically irrelevant differences between CHT and EMR data. The authors have pared down their claims to a level that fits within the scope of their paper. However there are still several mentions such as in the abstract, “CHT out-performed history taking by physicians” yet there is still no clinically relevant indication of this. While I understand the authors opinion that CHT was created by experts and is likely a better representation of an accurate chest pain history, there is no indication that clinical outcomes are at all effected by this. The authors should carefully re-read their manuscript to eliminate any indication that CHT outperforms EMR history in any manor that would improve clinical outcomes.

Overall, this version of the manuscript is much stronger. I think that while adding some relevant findings, the results section remains too long and verbose. Much of the “results” section should be moved to discussion instead.

Reviewer #2: (No Response)

Reviewer #3: The study authors have made several edits relating to the concerns from the first submission, however the manuscript as it requires further editing or rewriting to be at an acceptable level for publication.

Abstract – lacks objective (statistical) data to support its conclusions

Introduction – authors should remove their conclusions from the introduction – specifically “Our data shows that CHT records contain more complete and more accurate representations of patients' perceptions of their symptoms than patients' corresponding EHRs.” Please refer to the STROBE checklist if there is confusion on this point.

Methods – Gender and age data should be reported in the results section. Authors need to include a description of their statistical methods.

Results – The data on location of primary pain, location of radiated pain, setting of pain onset, and frequency/duration of pain, and data for associated symptoms, use of nitroglycerin, and dimensionality is very difficult to interpret. The descriptions are very long and would be best presented in table format, with limited narrative. Also, many values are reported as integers, where percentages would be better used. Finally, only raw data is presented; testing for statistical significance in the different outcomes between the EHR and CHT data sets would greatly strengthen the results.

Discussion - The authors should situate their findings in the context of prior research comparing EHRs and CHTs (ie. this is the first study, or consistent with prior studies, or contrary to prior studies) .

7. PLOS authors have the option to publish the peer review history of their article (what does this mean?). If published, this will include your full peer review and any attached files.

Reviewer #1: No

Reviewer #2: No

Reviewer #3: No

---

## [Author Response · Author response to Decision Letter 1]

23 Jul 2021

Replies to reviewers' objections.

Reviewer #1: In my first review of this paper the main issue I took was with a broad over-reaching conclusion based off of likely clinically irrelevant differences between CHT and EMR data. The authors have pared down their claims to a level that fits within the scope of their paper. 

However there are still several mentions such as in the abstract, “CHT out-performed history taking by physicians” yet there is still no clinically relevant indication of this. 

Reply: We removed the phrase and the complete sentence in which it appeared

----------------

While I understand the authors opinion that CHT was created by experts and is likely a better representation of an accurate chest pain history, there is no indication that clinical outcomes are at all effected by this. The authors should carefully re-read their manuscript to eliminate any indication that CHT outperforms EMR history in any manor that would improve clinical outcomes.

Reply: There is no mention in the manuscript that CHT will improve outcomes. 

---------------

Overall, this version of the manuscript is much stronger. I think that while adding some relevant findings, the results section remains too long and verbose. Much of the “results” section should be moved to discussion instead.

Reply: Results has no text not immediately relevant to the primary data. We see no material in Results that fit better in Discussion.

Reviewer #2: (No Response)

Reviewer #3: The study authors have made several edits relating to the concerns from the first submission, however the manuscript as it requires further editing or rewriting to be at an acceptable level for publication.

Abstract – lacks objective (statistical) data to support its conclusions

Reply: This item is addressed below.

-----------------

Introduction – authors should remove their conclusions from the introduction – specifically “Our data shows that CHT records contain more complete and more accurate representations of patients' perceptions of their symptoms than patients' corresponding EHRs.” Please refer to the STROBE checklist if there is confusion on this point.

Reply: We removed this sentence from the Introduction.

Methods – Gender and age data should be reported in the results section. Authors need to include a description of their statistical methods.

Reply: Instructions indicated that this data was to be included in Methods.

-----------------

Results – 1. The data on location of primary pain, location of radiated pain, setting of pain onset, and frequency/duration of pain, and data for associated symptoms, use of nitroglycerin, and dimensionality is very difficult to interpret. 

The descriptions are very long and would be best presented in table format, with limited narrative.

Reply: At the reviewer's suggestion, we added a table (Table 2 in the revised ms) for the data on sites of primary pain. We added a table (Table 4) for data for onset of pain during physical activity or emotional upset. (Data for other variables already is in Tables.)

-------------

Also, many values are reported as integers, where percentages would be better used. 

Reply: The differences between findings in EHR and CHT data are large for the absolute findings for different variables. Absolute values for what was found enables the reader easily to replicate the statistical test data should they want to do so. We note too that there is more than 1 way to calculate per cent of some number of records because of the large numbers of missing data elements in EHRs. The reader easily can calculate any per cent value that might interest them.

---------------

Finally, only raw data is presented; testing for statistical significance in the different outcomes between the EHR and CHT data sets would greatly strengthen the results.

Reply: We have added statistical analyses for all variables. We have added a brief section in Methods describing how statistical significance was determined. 

----------------

Discussion - The authors should situate their findings in the context of prior research comparing EHRs and CHTs (ie. this is the first study, or consistent with prior studies, or contrary to prior studies) .

Reply: We have added a sentence that contrasts the current work with prior work on this subject.

---

## [Decision Letter · Decision Letter 2]

8 Sep 2021

Computerized History-Taking Improves Data Quality for Clinical Decision-Making. Comparison of EHR and Computer-Acquired History Data in Patients with Chest Pain.

PONE-D-21-11324R2

Dear Dr. Zakim,

We’re pleased to inform you that your manuscript has been judged scientifically suitable for publication and will be formally accepted for publication once it meets all outstanding technical requirements.

Kind regards,

Amit Bahl

Academic Editor

PLOS ONE

Additional Editor Comments (optional):

Reviewers' comments:

Reviewer's Responses to Questions

**Comments to the Author**

1. If the authors have adequately addressed your comments raised in a previous round of review and you feel that this manuscript is now acceptable for publication, you may indicate that here to bypass the “Comments to the Author” section, enter your conflict of interest statement in the “Confidential to Editor” section, and submit your "Accept" recommendation.

Reviewer #1: All comments have been addressed

Reviewer #3: All comments have been addressed

2. Is the manuscript technically sound, and do the data support the conclusions?

Reviewer #1: Yes

Reviewer #3: Yes

3. Has the statistical analysis been performed appropriately and rigorously? 

Reviewer #1: Yes

Reviewer #3: Yes

4. Have the authors made all data underlying the findings in their manuscript fully available?

Reviewer #1: Yes

Reviewer #3: Yes

5. Is the manuscript presented in an intelligible fashion and written in standard English?

Reviewer #1: Yes

Reviewer #3: Yes

6. Review Comments to the Author

Reviewer #1: (No Response)

Reviewer #3: The authors have satisfactorily addressed the issues I highlighted in the last review of the manuscript. I thank them for this interesting and important study.

7. PLOS authors have the option to publish the peer review history of their article (what does this mean?). If published, this will include your full peer review and any attached files.

Reviewer #1: No

Reviewer #3: No

---

## [Editor Report · Acceptance letter]

13 Sep 2021

PONE-D-21-11324R2 

Computerized History-Taking Improves Data Quality for Clinical Decision-Making. Comparison of EHR and Computer-Acquired History Data in Patients with Chest Pain. 

Dear Dr. Zakim:

I'm pleased to inform you that your manuscript has been deemed suitable for publication in PLOS ONE. Congratulations! Your manuscript is now with our production department. 

Kind regards, 

on behalf of

Dr. Amit Bahl 

Academic Editor

PLOS ONE